# Digital home monitoring for capturing daily fluctuation of symptoms; a longitudinal repeated measures study: Long Covid Multi-disciplinary Consortium to Optimise Treatments and Services across the NHS (a LOCOMOTION study)

Maedeh Mansoubi ![ORCID] [1,2,3] Joanna Dawes,[2] Aishwarya Bhatia,[2]
Himanshu Vashisht,[4] Johnny Collett,[5] Darren C Greenwood ![ORCID] [6] Leisle Ezekiel,[7]
Daryl O'Connor ![ORCID] [8] Phaedra Leveridge ![ORCID] [3] Clare Rayner,[9] Flo Read,[10]
Manoj Sivan ![ORCID] [11] Ian Tuckerbell,[9] Tomas Ward,[12] Brendan Delaney,[13]
Willie Muhlhausen,[4] Locomotion consortium, Helen Dawes ![ORCID] [1,2,3]

For numbered affiliations see end of article.

**Correspondence to**
Professor Helen Dawes;
h.dawes@exeter.ac.uk

## ABSTRACT

**Introduction** A substantial proportion of COVID-19 survivors continue to have symptoms more than 3 months after infection, especially of those who required medical intervention. Lasting symptoms are wide-ranging, and presentation varies between individuals and fluctuates within an individual. Improved understanding of undulation in symptoms and triggers may improve efficacy of healthcare providers and enable individuals to better self-manage their Long Covid. We present a protocol where we aim to develop and examine the feasibility and usability of digital home monitoring for capturing daily fluctuation of symptoms in individuals with Long Covid and provide data to facilitate a personalised approach to the classification and management of Long Covid symptoms.

**Methods and analysis** This study is a longitudinal prospective cohort study of adults with Long Covid accessing 10 National Health Service (NHS) rehabilitation services in the UK. We aim to recruit 400 people from participating NHS sites. At referral to study, 6 weeks and 12 weeks, participants will complete demographic data (referral to study) and clinical outcome measures, including ecological momentary assessment (EMA) using personal mobile devices. EMA items are adapted from the COVID-19 Yorkshire Rehabilitation Scale items and include self-reported activities, symptoms and psychological factors. Passive activity data will be collected through wrist-worn sensors. We will use latent class growth models to identify trajectories of experience, potential phenotypes defined by co-occurrence of symptoms and inter-relationships between stressors, symptoms and participation in daily activities. We anticipate that n=300 participants provide 80% power to detect a 20% improvement in fatigue over 12 weeks in one class of patients relative to another.

## STRENGTHS AND LIMITATIONS OF THIS STUDY

⇒ Recruitment of participants across multiple National Health Service Long Covid clinics across England.
⇒ Readily available technology is used to monitor symptoms in real time in the context of daily life.
⇒ The reliability and validity of ecological momentary assessment (EMA) constructs are limited.
⇒ EMA completion rates and participant assessment reactivity are limited.
⇒ The longitudinal and remote monitoring aspect of the study means there is a risk of some participants not engaging with the study and participant attrition.

**Ethics and dissemination** The study was approved by the Yorkshire & The Humber—Bradford Leeds Research Ethics Committee (ref: 21/YH/0276). Findings will be disseminated in peer-reviewed publications and presented at conferences.

**Trial registration number** ISRCTN15022307.

## INTRODUCTION

Since the start of the COVID-19 pandemic in 2019, over 480 million people across the globe have had confirmed COVID-19 infection.[1] A substantial proportion of COVID-19 survivors will continue to have symptoms more than 3 months after infection, especially, but not restricted to, those requiring medical intervention for the infection.[2–5] The term Long Covid is used to define the long-lasting symptoms of COVID-19 and includes ongoing symptomatic COVID-19 (from 4 to 12 weeks)

and post-COVID-19 syndrome (12 weeks or more).[6] In the UK alone, an estimated 1.5 million[5] and over 100 million people globally[2] are living with self-reported Long Covid.[5]

The presentation of Long Covid is variable with a wide range of symptoms, which most commonly include general weakness, malaise, fatigue, breathlessness and impaired concentration.[7–9] Long Covid has been described as having 'undulating symptoms' where individuals may experience feeling well but then have a reoccurrence of symptoms, sometimes in response to triggers such as stress or physical activity.[10] The need for a greater degree of definitional stringency and clearer classification of symptom incidences, number, severity, trajectory and duration for clinical trials has been highlighted.[11] To address this, and a need to improve access to multidisciplinary healthcare for the large numbers of people reporting Long Covid,[10] we propose exploring the use of remote monitoring methods with readily available technology as a potential tool to assist patients and healthcare professionals in the clearer classification of symptoms and a better understanding of symptom management and help with recovery.[12 13]

There are currently no validated monitoring systems to classify and regularly check a patient's symptoms and report them to doctors and other healthcare professionals. A recent systematic review including 272 articles demonstrated that remote patient monitoring (RPM) methods could improve patient support and the efficiency of the therapy.[14] According to this review, smartphone applications and wireless devices are the most common RPM methods.[14] Given the lack of evidence on a valid and feasible method to support patients and health systems with COVID symptom RPM and management, this work package of the LOCOMOTION study[15] aims to develop and examine the feasibility and usability of home monitoring (using smartphone applications and wearable sensors) for capturing the daily fluctuation of symptoms in real time, recognising adverse triggers across patients from a range of demographic groups and aid self-management.

The primary objectives of the study are:

1. To quantify the extent to which total (per day and week) and specific (at each time point) cognitive, physical, and social activities, sleep and rest predict subsequent incidence, severity, number and duration of Long Covid symptoms, including postexertional malaise (PEM). By total activities, this project refers to the overall engagement in cognitive, physical and social activities over a given period. Specific activities encompass the individual types of cognitive, physical and social activities conducted at each specific time point.

2. To identify different groups or classes of people who respond to physical or mental effort (based on their response to questionnaires, app surveys and accelerometer result) differently in regard to PEM, other symptom incidences, duration, number and severity. Responding to physical or mental effort relates to the participant's response or experience following engagement in activities that require physical or mental exertion. This can include fatigue, symptom exacerbation or any other subjective or objective measures of response to effort.

The secondary objectives of the study are to:

3. Quantify the extent to which the prevalence or severity of specific symptoms predicts subsequent activity.

4. Quantify the relationship between sleep duration and quality with Long Covid symptoms.

5. Explore the relationship between different component contributions to the overall level of physical activity (frequency, intensity, recovery, duration) and subsequent symptoms.

6. Explore the associations between different timing or degree of exposure to (cognitive, physical and social) activities in terms of cumulative exposure (eg, multiple triggers in a short space of time or repeated triggers over the day), along with the time-lagged onset of symptoms or exacerbation of symptom severity.

7. Determine the extent of the relationship between physiological variables and activity and symptom severity for different groups of activities and symptoms (substudy 1).

8. Describe the stability of the activity, symptoms and the interrelationship in people with Long Covid.

9. Describe the variability in activities, dose of activities and symptom severity.

10. Explore the moderating effects of living with physical health and/or mental health conditions (including depression and anxiety) on the relationships between the triggers and symptoms of Long Covid (as outlined in objectives 1 and 2).

11. Describe the concurrent validity of components of the home monitoring system (substudy 1).

12. Investigate usability and acceptability of home monitoring system and explore patient experiences of Long Covid (substudy 2).

## METHODS AND ANALYSIS

This is workstream 2.1 of the LOCOMOTION Health Service Study (trial registration number ISRCTN15022307).[15] The objective of this workstream is to specifically focus on the remote monitoring of individuals with Long Covid. It aims to assess the effectiveness of remote monitoring techniques in tracking and managing the symptoms of Long Covid. The study follows the Strengthening the Reporting of Observational Studies in Epidemiology (STROBE) guidelines (https://www.equator-network.org/reporting-guidelines/strobe/).

### Design and setting

The study planned start date is February 2022, and the anticipated end date is September 2023. Using a longitudinal study design with repeated measures and a nested substudy, we aim to recruit a purposive sample of patients with (self-reported/general practitioner diagnosed) Long Covid who have been referred to one of the ten

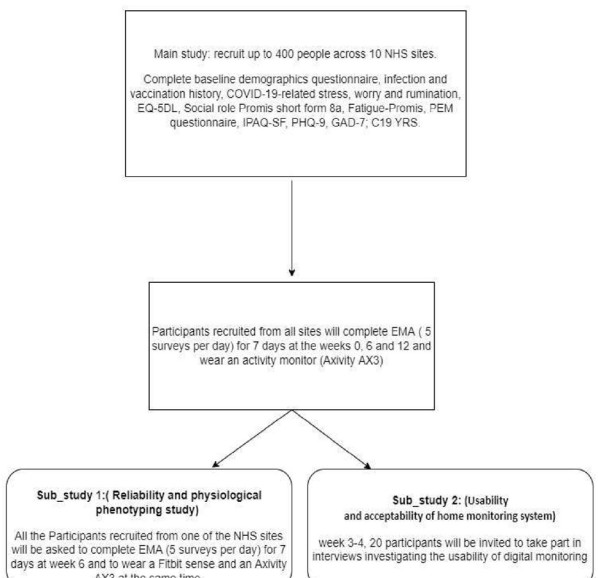

**Figure 1** Main and substudies flow chart. C19 YRS, COVID-19 Yorkshire Rehabilitation Scale; EMA, ecological momentary assessment; EQ-5D, EuroQol-5D; GAD-7, Generalised Anxiety Disorder Assessment; NHS, National Health Service; PEM, postexertional malaise; PHQ-9, 9-question Patient Health Questionnaire.

participating National Health Service (NHS) rehabilitation clinics (see figure 1). We will recruit 400 participants with the anticipation of a 25% drop-out, giving 300 patients (40 per site) using purposive sampling.

At referral to study, at 6 weeks and 12 weeks, participants will complete demographic data (on referral to study) and clinical outcome measures, including ecological momentary assessment (EMA) and passive activity data over 7 days using personal mobile devices. EMA symptoms items are adapted from the COVID-19 Yorkshire Rehabilitation Scale (C19-YRS) items and include self-reported activities, symptoms and psychological factors. Passive activity data will be collected through wrist-worn sensors. EMA consists of sampling an individual's experiences and subjective states as they occur in real-time in the individual's natural environment.[16] There will be two embedded substudies. Substudy 1 will include additional home digital physiological measures of heart rate variability (HRV) to test biofeedback (improving symptoms by increasing HRV). Substudy 2 will use qualitative interviews to assess the usability and acceptability of using home monitoring (via smartphones) of triggers and symptoms of Long Covid at clinic appointments and explore patient experiences of Long Covid.

### Inclusion criteria
We will include all adults who are 18 years and over, able to participate in the study (ie, able to complete the EMA and use the wearable technology), able to give informed consent and have English language proficiency to the level of conducting health consultations. Participants will not be included or excluded based on past or current illness or medication at the time of the presumed COVID-19

infection; however, they will be asked to describe these at the start of this study.

### Exclusion criteria
Participants will be excluded who are under 18 years old, unable to use the EMA app technology or use the wearable technology, unable to understand the language used in the EMA or are known to be pregnant. We will also exclude those with a known previous diagnosis of dementia or cognitive impairment that would prevent ability to participate in the use of the EMA or wearable technologies.

### Main study
The main LOCOMOTION study, encompassing all workstreams, aims to recruit a total of 5000 participants. Within this comprehensive study, workstream 2.1 specifically focuses on recruiting 400 individuals with Long Covid. This dedicated workstream aims to explore and evaluate remote monitoring techniques for managing and tracking the symptoms of Long Covid. By recruiting this specific cohort, we aim to gather valuable insights and data to enhance the understanding and treatment of Long Covid within the broader LOCOMOTION study. With broad assumptions, based on pilot data from our rehabilitation clinics, we anticipate n=300 participants would provide approximately 80% power to detect 20% improvement in fatigue over 12 weeks in one class of patients relative to another, for example, a stable symptom group with fatigue scores of 5 at each time point, compared with a group gradually improving from 5 to 4, with within-person correlation 0.7, residual variance 2.0 and 30% with the symptom pattern. We anticipate up to 25% drop-out across follow-up, so we initially aim to recruit up to n=400.

### Substudy 1: (reliability and physiological phenotyping study of home monitoring system)
This study will invite 50 participants, allowing for dropouts and leaving a sample size of approximately 30 participants to ensure meeting central limit theorem criteria for reliability analysis and between 25 and 50 for meeting feasibility criteria.[17]

### Substudy 2: (usability and acceptability of home monitoring system)
This study will invite 20 patients who have taken part in the main monitoring study to be interviewed and 10 clinicians working in one of the collaborating Long Covid clinics. This sample size should ensure sufficient 'information power' to address the research questions.[18]

### Recruitment
Each of the 10 participating Long Covid clinics has a clinical research lead who is responsible for inviting patients to participate in the study on referral. Potential participants will be notified of the opportunity to take part in the study by the clinical research lead at each site and provided with a link to an online form to register their

expression of interest and check that they meet the study inclusion criteria. Intake will be capped when recruitment has achieved the study sample size, and further applicants will receive an email response that the study is closed but that they will be contacted again when it is open.

## Data collection

### Baseline and follow-up data collection

Baseline and follow-up measures will be collected using the online data capture platform REDCap. On entry to the study, participants will also complete the C19-YRS and provide information about their demographics (age, height, weight, sex, ethnicity, postcode, date of birth, marital status, housing, education, employment, smoking, disability, medical history, drug history, covid history and care). Additionally, at weeks 1, 6 and 12e, participants will complete the C19-YRS[19] and provide information about their physical activity levels, fatigue, quality of life and participation, well-being, anxiety and rumination, worry, depression and PEM.

### Wearables for home monitoring of fluctuation of symptoms

At referral to the study, at 6 weeks and 12 weeks, participants will be asked to perform EMA over 7 days using personal mobile devices and wearable sensors and accelerometers. The EMA is delivered using the AthenaCX platform, a secure and General Data Protection Regulation (GDPR)-compliant research survey platform[20] which works on Apple and android phones. The EMA includes five time-contingent surveys (sent between 8.45 and 21.15 hours) of self-reported activities, symptoms and psychological factors (see online supplemental appendix 1). EMA questions were adapted from the C19-YRS items to enable home monitoring of symptom fluctuation and trigger identification, and from items developed for the UK COVID-19 Mental Health and Well-being study.[21] Pilot work has demonstrated that this approach is feasible and acceptable to Long Covid patients.

### Accelerometer sensor

The daily amount of physical activity and sedentary time will be objectively assessed using the Axivity AX3, a 'wrist-worn' tri-axial accelerometer designed by Open Lab, Newcastle University, UK. AX3 data will be downloaded using OMGUI software (open movement (V.1.0.0.37)). The Axivity AX3 is a valid and reliable method for the measurement of physical activity and sedentary behaviour.[22]

### Substudy 1: (reliability and physiological phenotyping study of home monitoring system)

Reliability and physiological phenotyping study of 50 patients in Oxford to include additional home digital physiological measures HRV.

Method: Fifty patients from the clinic in Oxford will be invited to participate in the following voluntarily and will be able to opt-in or opt-out of any of the components at each testing time point: digitally and remotely monitoring physiological measures via a wrist-worn or a chest strap heart rate sensor, based on participants' preference and capability to measure heart rate and HRV. This aims to explore the potential to use measuring heart rate and HRV as feedback for clinical purposes.

### Substudy 2: process evaluation (usability and acceptability of home monitoring system)

Usability, acceptability and codesign qualitative study with 20 patients and ten clinicians and a clinician survey to assess the usability and acceptability of using home monitoring (via smartphones) of triggers and symptoms of Long Covid at clinic appointments and explore patient experiences of Long Covid.

Method: Substudy 2 will be a codesign study between 20 patients and 10 clinicians. It will use a survey to explore symptom behaviour and establish the utility of prospective symptom fluctuation and triggers for self-management. We will conduct a short debrief interview with up to 20 participants during either week three or week four of their study. This will be conducted by phone/video call and recorded. The interview will be structured around the usability dimensions of efficiency, effectiveness and satisfaction. In particular, the interview will ask about both the apps and using them in daily life, any issues identified and any suggestions for improvement to the app or the study design. The interview will include a 'would you recommend to someone you know?' question and explore reasons for this. The interview will also include a think-aloud component in which the interviewer asks the participant to go through a data entry cycle and describe what they are thinking or doing as they go along. Towards the end of the interview, the interviewee will be asked to complete the 10-item System Usability Scale. Participants will also be interviewed about the experiences of Long Covid.

For clinicians, the acceptability and usability assessment will focus on content, design, language and adaptability of daily monitoring information in the clinic environment and explore views on the usefulness of this approach to support Long Covid patients. Semistructured interviews will be conducted either face to face or via the secure web video conferencing system.

### Data cleaning

Reported values more than 3 SDs from the mean will be double-checked with original data sources. Observations with studentised residuals >3, Cook's distance greater than 1 and leverage of h>0.3 will be used to detect influential data points in subsequent models. No data will be excluded without evidence of error or implausibility, other than through sensitivity analyses.

### Descriptive statistics

Patient characteristics (demographics, clinical history, exposures and outcomes) will be tabulated overall, by sex, by 2–5 broad age bands derived from the data to provide similar numbers in each age group, and by 2–5 fatigue bands, similarly derived, with sensible rounding

of thresholds (eg, to nearest 5 years or to nearest 1 point on scale).

Visual Analogue Scales and measures will be presented as median and IQR or mean and SD, dependent on the underlying distributions of the scores.

The characteristics of consenting participants will be compared with the characteristics of all clinic patients, to investigate whether the participating patients differ systematically from those not participating. Characteristics of participants with incomplete data will be compared with those with complete data.

## Statistical modelling
### Primary objective 1
Multilevel multivariate vector autoregressive (VAR) modelling will be used to investigate the associations between C19-YRS symptom scores, and the activities reported in the EMA, taking account of the hierarchical/nested multilevel structure of the data (objective 1). We will construct multilevel multivariate VAR models within the dynamic structural equation model module in Mplus V.8.6 or WinBUGS V.1.4 to partition total variability across individuals and time points into within-person and between-person components. We anticipate using non-informative priors and two independent Markov chain Monte Carlo chains, allowed to burn in until distributions are stable.

The within-person component will be modelled using a VAR time-series model of order 1. This model will be used to predict the outcomes at one-time point using all predictor variables at the previous time point. This will allow estimation of the within-person associations between symptom outcomes and potential triggers, the autoregressive effects that the severity of the outcome variable has on itself at the next time point, and the cross-lagged effects of one variable on another variable at the next time point. Both autoregressive and cross-lagged parameters will be modelled as random effects that vary across individuals. We anticipate fitting both random intercepts and slopes but assuming these are independent of each other. Robustness to this assumption will be assessed in sensitivity analyses, where sufficient df and complexity of models allow.

We will estimate the association of between-individual differences in the random-effects with variation in other variables measured at the between-person level, for example, whether change symptoms over time are associated with within-person dynamics between symptom severity and potential triggers such as activities and effort, by specifying the change in symptom severity as the outcome variable and the random effects of the cross-lagged and autoregressive parameters as predictors. All variables and parameters will be centred, standardised or detrended where appropriate to facilitate correct analysis and interpretation.

To allow for both shorter-term and longer-term lag effects between triggers and outcomes, analyses will be conducted first with each EMA as the time point (eg, to compare mental effort at one assessment with fatigue at the next), then repeated using the whole day as the time point, aggregating over the 6 EMAs that day (eg, to compare impact of physical effort one day on symptoms the next day). This will also be used for any outcomes only measured once in a day.

The same methods will be applied to secondary objective 3 (symptoms predicting subsequent activity), objective 4 (timing, duration and quality of sleep), objective 5 (components of physical activity), objective 6 (timing and cumulation of activities) and in part to objectives 7 (HRV and physiological phenotyping) and 8 (stability of association).

### Primary objective 2
Latent class growth models will be used to identify different types of symptom trajectories experienced by those living with Long Covid, potential phenotypes defined by co-occurrence of symptoms and any interrelationships between triggers, symptoms and participation in daily activities, personal care such as washing or dressing, usual activities, problems walking about, problems communicating with others, problems with concentration or remembering things (objective 2). The extent of these between-person and within-person day-to-day fluctuations will be quantified, and the potential for characteristics to predict subsequent improvement over time. The number of classes will be based on information criteria (eg, Bayesian information criterion (BIC) and Akaike information criterion (AIC)) and clarity of interpretation of the domains identified.

Functional data analysis will also be used, where appropriate, to characterise features of long-term symptom trajectories, such as variation and speed of change. The extent of these between-person and within-person day-to-day fluctuations will be quantified, and the potential for characteristics to predict subsequent improvement over time.

These methods will also be applied in part to objectives 7 (HRV and physiological phenotyping) and 8 (stability of association).

All statistical analyses will be conducted using SPSS V.28, R, Stata V.17 and MPlus V.8.6 or WinBUGS V.1.4.

## Bias
### Potential confounding
Potential confounders will be identified based on directed acyclic graphs, but as the aim is prediction, we will also include potential competing exposures in models.

### Missing and incomplete data
The latent class growth models use mixed-effects modelling that allows for partially incomplete repeated measures across participants.

### Exclusions, sensitivity analyses and subgroup analyses
Sensitivity analyses will be conducted to assess robustness to different types of trajectories for men vs women, hospitalised versus non-hospitalised, individuals with existing

mental health conditions versus individuals without existing mental health conditions, different waves of acute infection as a proxy for variant, broad age band.

We have previously described sensitivity analysis assessing robustness of the models to the timeframe used for the VAR analysis, that is, EMA or summary measure over the whole day We will also assess robustness of the model to second order time-lags, in addition to the first order time-lags, where the data allow.

We assume independence of random intercepts and slopes but will assess robustness to this correlation structure where the data allow.

As sensitivity analysis to the effects of incomplete data, we will repeat excluding any days with less than two EMAs.

### Thematic analysis

Thematic content analysis of interview transcripts will be used to investigate the usability and perceived usefulness of the digital measurement tools (by clinicians and patients) to support self-management and rehabilitation. It will be also used to explore participants' experiences of Long Covid triggers and symptoms. All qualitative data collection for the process evaluation will be carried out by a research team member with qualitative research experience. Interviews will be transcribed verbatim and analysed using thematic analysis methods which involves familiarisation with the interview, coding, developing and applying an analytical framework, charting data into the analytical framework for analysis.[23] Interview transcripts will be coded using NVivo software (or similar software). Each interview will be independently coded by two reviewers. After coding four transcripts, reviewers will compare codes and discrepancies will be discussed and resolved prior to coding the remaining transcripts. The interim analysis will be conducted following an initial sample of 150 patients to determine whether saturation of themes had been reached. Mean (±SD) or median and IQR values will be used as appropriate to summarise participants' demographic data.

Interviews will be analysed thematically initially to explore usability factors. Due to the focus of this process evaluation, the personal documentation will be initially subjected to content analysis—with a more detailed thematic analysis applied if required. Data will be reviewed in light of any contradictions and will guide a member-checking process with participants, which will be conducted prior to the write-up and dissemination of findings. Findings will be separated into the process and measure ready for reporting, but with consideration being given to where/if the doing of any process is a major contributor to the outcomes or perceived experience of participants. At this point, we will explore possible mechanisms for any observed effects, to generate change objectives and behavioural outcomes.

### Public patient involvement and engagement

The team have regular public patient involvement (PPI) meetings, and the study was designed in consultation with Long Covid patients on how COVID-19 has disproportionately affected different communities. The LOCOMOTION study has a seven-member core PPI advisory group that includes different cultural, ethnic and socioeconomic groups. They have contributed to the proposal research planning meetings and met separately to examine and develop the research aim, objectives and questions ensuring these align with the key research priorities of patients with Long Covid and represent different patient needs.

### Ethics and dissemination

The study was approved by the Yorkshire & The Humber—Bradford Leeds Research Ethics Committee (ref: 21/YH/0276) and conforms with the Declaration of Helsinki.

Participants consent to participate following an e-consent process. The e-consent form will be implemented through a secure web-based online survey accessible by computer, mobile phone or tablet. The e-consent process also includes questions about the person's understanding of the study based on the Brief Assessment of Capacity to Consent;[24] participants need to answer at least six of eight questions correctly to be eligible for the study. This process queries the person's understanding of the consent form and has been used in over 200 research studies worldwide. A parallel consent process using telephones is available for those participants who are not able to use, or do not have access to the internet. Potential participants will be telephoned when the clinical research lead indicates this is the preferred mode of contact and communication to the research team. A call will be set up for one of the research teams to provide them with information. If the potential participant is interested, a second call will elicit their consent over the phone using the same script as e-consent and conduct the screening process using the same questionnaire. The questions on the capacity to consent quiz will be asked orally.

Findings will be disseminated through manuscript publications in peer-reviewed journals and conference presentations.

## DISCUSSION

The overarching aim of this study is to optimise the classification and self-management of Long Covid by tracking symptoms in real time in the context of daily life, and by using a data-driven approach to identify adverse triggers of symptoms and phenotypes of Long Covid. These findings also aim to extend knowledge of the trajectory of Long Covid, particularly interactions between Long Covid symptoms and activity-related behaviours. Understanding the relationship between Long Covid symptoms and everyday activity is essential for supporting effective self-management of Long Covid, particularly on achieving balance of activity, priority of activity in order to manage social, family and work commitments. The study findings will help facilitate a personalised approach to the

management of Long Covid based on ecologically valid self-report data.

Additionally, we will evaluate the usability of digital home monitoring by patients and clinicians. Taken together, the findings of these studies will help support the potential development of a digital home monitoring, self-management platform for use with Long Covid patients. It is hoped that a future home monitoring platform will enable the provision of cost and time-effective care.

## Strengths and limitations

### Strengths

Key strengths of this study are the planned recruitment of participants through multiple NHS Long Covid clinics across England, and the use of readily available technology and real-time monitoring of symptoms in the context of daily life. These aspects of the study design increase the generalisability and ecological validity of the study findings.

### Limitations

There are several known methodological limitations to EMA: the reliability and validity of EMA constructs, EMA completion rates and participant assessment reactivity.[25] In addition, the longitudinal and remote monitoring aspect of the study means there is a risk of some participants not engaging with the study, and of participant attrition. We will use several strategies to minimise these limitations. First, we optimised the face and construct validity of the EMA questions by using single constructs items from the C19-YRS[19] and PEM.[26] In addition, the EMA was tested by people with Long Covid and clinicians with expertise in Long Covid to check the clarity and relevance of questions. We also tested the EMA schedule with people with Long Covid to ensure the schedule was acceptable and to optimise completion rates. We planned bursts of assessment (7 days repeated three times) rather than extended periods, thereby reducing the burden on the participant. Finally, we recognise the need to support participants through the first stages of using the EMA app; therefore, each participant will have an initial phone call from a member of the research team and subsequently have ongoing contact with the team via email or telephone.

**Author affiliations**
[1]NIHR Exeter Biomedical Research Center, Medical School, Faculty of Health and Life sciences, University of Exeter, Exeter, UK
[2]Medical School, University of Exeter, Exeter, UK
[3]Department of Public Health and Sport Sciences, University of Exeter, Exeter, UK
[4]In The Wild Research Limited, Dublin, Ireland
[5]Department of Sport, Health and Social Work, Oxford Brookes University, Oxford, UK
[6]Academic Department of Rehabilitation Medicine, University of Leeds, Leeds, UK
[7]School of Health Sciences, University of Southampton, Southampton, UK
[8]School of Psychology, University of Leeds, Leeds, UK
[9]Patient Advisory Group (PAG) Representative, Leeds, UK
[10]Department of Health and Community Sciences, University of Exeter, Exeter, UK
[11]Faculty of Medicine and Health, School of Medicine, University of Leeds, Leeds, UK
[12]Insight SFI Research Centre for Data Analytics, Dublin City University, Dublin, Ireland
[13]Faculty of Medicine, Department of Surgery & Cancer, Imperial College, London, UK

**Acknowledgements** This study was supported by the National Institute for Health and Care Research. HD is funded and supported, and MM's work is supported by the National Institute for Health and Care Research Exeter Biomedical Research Centre. JC is funded by the NIHR Oxford Health Biomedical Research Centre.

**Collaborators** Locomotion consortium, Locomotion consortiumLOCOMOTION consortium: Nawar Diar Bakerly, Mauricio Barahona, Alexander Casson, Jonathan Clarke, Vasa Curcin, Helen Davies, Carlos Echevarria, Sarah Elkin, Rachael Evans, Zaccheus Falope, Ben Glampson, Trisha Greenhalgh, Stephen Halpin, Mike Horton, Joseph Kwon, Simon de Lusignan, Gayathri Delanerolle, Erik Mayer, Harsha Master, Ruairidh Milne, Ghazala Mir, Jacqui Morris, Amy Parkin, Stavros Petrou, Anton Pick, Nick Preston, Amy Rebane, Emma Tucker, Ana Belen Espinosa Gonzalez, Sareeta Baley, Annette Rolls, Emily Bullock, Megan Ball, Shehnaz Bashir, Joanne Elwin, Denys Prociuk, Iram Qureshi, Samantha Jones.

**Contributors** All authors meet the ICMJE statement. All the below contributed to study design, drafting editing or revising paper, and have given final approval and agreed to accountability. MM, JD, JC, LE and HD, BD, DCG, DO'C, CR, MS, IT-B, TW, HV, AB, FR, PL and WM. LOCOMOTION consortium authors contributed to the protocol design.

**Funding** This project has been funded by NIHR, grant reference number: COV-LT2-0016.

**Disclaimer** The views expressed are those of the author(s) and not necessarily those of the NIHR or the Department of Health and Social Care.

**Competing interests** None declared.

**Patient and public involvement** Patients and/or the public were involved in the design, or conduct, or reporting, or dissemination plans of this research. Refer to the Methods section for further details.

**Patient consent for publication** Not applicable.

**Provenance and peer review** Not commissioned; externally peer reviewed.

**ORCID iDs**
Maedeh Mansoubi http://orcid.org/0000-0002-8829-2217
Darren C Greenwood http://orcid.org/0000-0001-7035-3096
Daryl O'Connor http://orcid.org/0000-0003-4117-4093
Phaedra Leveridge http://orcid.org/0000-0002-3602-2231
Manoj Sivan http://orcid.org/0000-0002-0334-2968
Helen Dawes http://orcid.org/0000-0002-2933-5213

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
