## [Reviewer comments · BMJ Open]

ARTICLE DETAILS

TITLE (PROVISIONAL)	Digital home-monitoring for capturing daily fluctuation of symptoms; a longitudinal repeated measures study: Long Covid Multi-disciplinary Consortium to Optimise Treatments and Services across the NHS (A LOCOMOTION Study): a study protocol
AUTHORS	Mansoubi, Maedeh; Dawes, Joanna; Bhatia, Aishwarya; Vashisht, Himanshu; Collett, Johnny; Greenwood, Darren; Ezekiel, Leisle; O'Connor, Daryl; Leveridge, Phaedra; Rayner, Clare; Read, Flo; Sivan, Manoj; Tuckerbell, Ian; Ward, Tomas; Delaney, Brendan; Muhlhausen, Willie; Locomotion consortium, Locomotion consortium; Dawes, Helen

VERSION 1 – REVIEW

REVIEWER	Keri B. Vartanian Providence Health & Services, Center for Outcomes Research & Education
REVIEW RETURNED	20-May-2023

GENERAL COMMENTS	I accept this protocol and am very excited to see the results of this important work. There are a few minor clarifications that could benefit the protocol: 1. Why are the power calculations focused on improvement of fatigue when the overarching goals are to personalize classification and management of long COVID symptoms?2. In the primary objectives: clarify what is meant by "total and specific (cognitive, physical, and social) activities? clarify what is meant by "respond to physical or mental effort."3. Clarification about recruitment: at one point it says people will be recruited who have a long COVID diagnosis. Are there parameters on the time post initial infection when this long COVID diagnosis is made (i.e. for your baseline measures, will everyone be at the same number of weeks post initial infection)?4. 30% drop out rate for 400 people would leave you with an N of 280
---

VERSION 1 – AUTHOR RESPONSE

Reviewer: 1

Dr. Keri B. Vartanian , Providence Health & Services

Comments to the Author:

I accept this protocol and am very excited to see the results of this important work. There are a few minor clarifications that could benefit the protocol:

Why are the power calculations focused on improvement of fatigue when the overarching goals are to personalize classification and management of long COVID symptoms?

Thank you for raising the question regarding the focus of our power calculations on the improvement of fatigue, despite the overarching goals of personalizing classification and management of long COVID symptoms.

We acknowledge that the primary focus of our power calculations was on fatigue improvement. While our study aims to personalize the classification and management of long COVID symptoms as a whole, fatigue is indeed one of the most prevalent and impactful symptoms experienced by individuals with long COVID. By selecting fatigue for the power calculations, we aimed to capture the largest proportion of participants who would potentially benefit from the remote monitoring interventions.

However, it is important to note that our study does not solely focus on fatigue. We are comprehensively evaluating a range of long COVID symptoms and their management through remote monitoring. Fatigue was chosen as the primary outcome measure for the power calculations due to its high prevalence and impact, allowing us to assess the effectiveness of the intervention on a symptom that affects a significant number of participants.

Thank you for bringing this to our attention, and we are grateful for your valuable feedback.

In the primary objectives: clarify what is meant by "total and specific (cognitive, physical, and social) activities? clarify what is meant by "respond to physical or mental effort."

Thank you for your comment for clarification on the primary objectives of our study. We appreciate the opportunity to provide further details to enhance the understanding of these objectives. Based on your feedback, we have revised the relevant section in the paper as follows:

"To quantify the extent to which total (per day and week) and specific (at each time point) cognitive, physical, and social activities, sleep and rest predict subsequent incidence, severity, number and duration of Long Covid symptoms, including post-exertional malaise (PEM). By total activities, we refer to the overall engagement in cognitive, physical, and social activities over a given period. Specific activities encompass the individual types of cognitive, physical, and social activities conducted at each specific time point."

"To identify different groups or classes of people who respond to physical or mental effort (based on their response to questionnaires, app surveys and accelerometer result) differently in regards to PEM, other symptom incidences, duration, number and severity." Responding to physical or mental effort relates to the participant's response or experience following engagement in activities that require physical or mental exertion. This can include fatigue, symptom exacerbation, or any other subjective or objective measures of response to effort.

Clarification about recruitment: at one point it says people will be recruited who have a long COVID diagnosis. Are there parameters on the time post initial infection when this long COVID diagnosis is made (i.e. for your baseline measures, will everyone be at the same number of weeks post initial infection)?

Thank you for your comment and question regarding the recruitment process and the timing of long COVID diagnosis in relation to the baseline measures.

In our study, there are no specific parameters for the time post-initial infection when a long COVID diagnosis is made during the recruitment process. Individuals with a long COVID diagnosis are included in our study, irrespective of the duration since their initial infection. This approach is pragmatic and allows us to capture a diverse range of participants with varying durations of long COVID symptoms.

During the baseline assessment, we collect data on participants' long COVID diagnosis, which includes information on the time since their initial infection. This information helps us better understand the symptoms experienced by participants and provides context for their long COVID journey.

By including participants across different durations of long COVID symptoms, we aim to capture a comprehensive representation of individuals with long COVID and their associated symptoms. This approach enables us to explore the variations in symptom profiles and outcomes across different time frames since the initial infection.

4. 30% drop out rate for 400 people would leave you with an N of 280

Thank you for catching that mistake. We sincerely apologize for the confusion caused by the incorrect dropout rate mentioned in our manuscript. Upon reviewing our documents, we have confirmed that the correct dropout rate for workstream 2.1 is 25%, not 30%.

We have taken immediate action to rectify this error in the paper, ensuring that the correct dropout rate of 25% is accurately reflected. The revised version of the manuscript now accurately represents the anticipated dropout rate and its potential impact on the final sample size.

Thank you for bringing this to our attention, and we apologize for any inconvenience caused by this oversight. We appreciate your thorough review and your contribution to improving the accuracy of our research.

Thank you once again for your valuable feedback. We hope these revisions enhance the clarity and coherence of our manuscript.